# Diagnostics and Management of Pancreatic Cystic Lesions—New Techniques and Guidelines

**DOI:** 10.3390/jcm13164644

**Published:** 2024-08-08

**Authors:** Jagoda Rogowska, Jan Semeradt, Łukasz Durko, Ewa Małecka-Wojciesko

**Affiliations:** Department of Digestive Tract Diseases, Medical University of Lodz, 90-647 Lodz, Poland; jan.semeradt@stud.umed.lodz.pl (J.S.); lukasz.durko@umed.lodz.pl (Ł.D.); ewa.malecka-panas@umed.lodz.pl (E.M.-W.)

**Keywords:** endoscopic ultrasound, pancreatic cystic lesions, pancreatic cystic neoplasms, pancreatic cysts, pancreatic cyst guidelines

## Abstract

Pancreatic cystic lesions (PCLs) are increasingly diagnosed owing to the wide use of cross-sectional imaging techniques. Accurate identification of PCL categories is critical for determining the indications for surgical intervention or surveillance. The classification and management of PCLs rely on a comprehensive and interdisciplinary evaluation, integrating clinical data, imaging findings, and cyst fluid markers. EUS (endoscopic ultrasound) has become the widely used diagnostic tool for the differentiation of pancreatic cystic lesions, offering detailed evaluation of even small pancreatic lesions with high sensitivity and specificity. Additionally, endoscopic ultrasound–fine-needle aspiration enhances diagnostic capabilities through cytological analysis and the assessment of fluid viscosity, tumor glycoprotein concentration, amylase levels, and molecular scrutiny. These detailed insights play a pivotal role in improving the clinical prognosis and management of pancreatic neoplasms. This review will focus mainly on the latest recommendations for the differentiation, management, and treatment of pancreatic cystic lesions, highlighting their clinical significance.

## 1. Introduction

The prevalence of PCLs is increasing due to advancements in imaging technology, leading to higher detection rates of smaller lesions [1]. Techniques like contrast-enhanced ultrasound, high-resolution CT, and MRI/MRCP provide detailed cross-sectional images, improving sensitivity and accuracy. EUS, in particular, offers superior resolution and the ability to perform fine-needle aspiration for cyst fluid analysis, increasing diagnostic precision [2]. Several studies have documented this trend. For example, a systematic review noted that the global prevalence of PCLs detected using MRI ranges from 13% to 18%, with the prevalence increasing significantly in individuals over 50 years of age [2]. Pancreatic lesions constitute a diverse array of fluid-filled cavities within the pancreas, ranging from benign to potentially malignant entities, and they can be categorized into simple retention cysts, pseudocysts, and cystic neoplasms [3]. The increased prevalence of PCLs is largely a consequence of advancements in imaging modalities, such as computed tomography (CT), magnetic resonance imaging (MRI), and EUS. These high-resolution imaging techniques are now more frequently employed, leading to the incidental detection of PCLs in asymptomatic patients undergoing imaging for unrelated medical conditions [4,5]. According to the 2019 WHO classification of tumors of the digestive system [6], pancreatic cysts include pseudocysts, intraductal papillary mucinous neoplasms (IPMNs), serous cystic neoplasms (SCNs), mucinous cystic neoplasms (MCNs), and rarer types, like solid pseudopapillary epithelial neoplasms (SPENs). Accurate differentiation among these neoplasms impacts clinical decisions, such as the need for surgical intervention versus surveillance, the frequency of follow-ups, and the type of surgical approach required. For example, benign lesions like SCNs often require less aggressive management compared to MCNs and IPMNs, which have a higher potential for malignancy and may necessitate surgical resection. Early and accurate diagnosis thus helps in preventing overtreatment of benign lesions and ensuring timely intervention for potentially malignant or malignant cysts, thereby improving patient outcomes and reducing healthcare costs [2,4,5].

Non-neoplastic pancreatic cysts typically do not require surgical resection unless they cause symptoms [7]. In contrast, the management of neoplastic pancreatic cysts is more nuanced due to their varying degrees of malignant potential. Among neoplastic pancreatic cysts, those of mucinous, intraductal papillary mucinous, solid pseudopapillary, and neuroendocrine origin harbor the potential for malignancy, warranting consideration for resection. The decision revolves around weighing the risk of malignant transformation against the potential risks associated with either surgical intervention or continued surveillance. Conversely, serous cystic tumors are benign in nature, requiring resection only if symptomatic [2,8].

EUS-guided fine-needle aspiration (FNA), EUS-guided fine-needle-based confocal laser endomicroscopy (nCLE), and needle microforceps biopsy are promising techniques for differentiating between mucinous and non-mucinous cysts [9]. These advanced techniques offer significant advantages over traditional methods, enhancing diagnostic accuracy and aiding in therapeutic decisions. nCLE integrates EUS with confocal laser endomicroscopy (CLE) to achieve high-resolution, real-time in vivo microscopic imaging of tissues. Utilizing a fine needle equipped with a CLE probe, the technique allows for precise targeting and visualization of internal structures under ultrasound guidance. This advanced method is particularly valuable for the evaluation of pancreatic cysts, lymph nodes, and other deep-seated lesions, enhancing diagnostic accuracy and helping in therapeutic decisions [10]. Experts recommend EUS-FNA when it significantly impacts clinical management but advise against it if other imaging techniques provide a clear diagnosis or surgery is clearly needed. EUS-FNA helps differentiate benign from malignant lesions, confirm suspicious features, and plan treatment, preventing unnecessary surgery [11,12,13].

A recent meta-analysis showed that cyst fluid cytology had a sensitivity of 42% and a specificity of 99% for differentiating mucinous from non-mucinous pancreatic cystic neoplasms (PCNs). This means that cyst fluid cytology can correctly identify 42% of mucinous cysts (true positives) and correctly identify 99% of non-mucinous cysts (true negatives) [14]. EUS morphology alone has an accuracy ranging from 48% to 94% in distinguishing between mucinous and non-mucinous PCNs, with sensitivity (the true positive rate) ranging from 36% to 91% and specificity (the true negative rate) ranging from 45% to 81%. While cytology shows high specificity (83–100%), indicating a strong ability to correctly identify non-mucinous cysts, it has lower sensitivity (27–48%), indicating a weaker ability to correctly identify mucinous cysts [15,16,17]. The wide range of accuracy (48% to 94%) for EUS morphology in distinguishing between mucinous and non-mucinous pancreatic cystic neoplasms is due to several factors. Operator experience plays a significant role, as even experienced endosonographers show modest agreement in differentiating cyst types. The variability in cyst characteristics and the presence of specific features like septations and mural nodules also impact accuracy. Additionally, the diagnostic performance of EUS improves when combined with fine-needle aspiration (FNA) for cytology and biochemical analysis of cyst fluid [18,19]. EUS-FNA is generally safe, with a low complication risk of 0.1%, though it is relatively contraindicated in patients with a high risk of bleeding (INR > 1.2, platelets < 100,000, or the use of dual antiplatelet therapy) or a distance between the cyst and the transducer >10 mm [11,20]. Patients receiving antiplatelet or anticoagulant drugs should discontinue their medication before the procedure. For patients with a high risk of thromboembolic events, bridge therapy with low-molecular-weight heparin should be considered. If the patient cannot safely discontinue their treatment and the risk of stopping anticoagulation surpasses the risk of FNA-induced bleeding, it may be reasonable to attempt EUS-FNA using a small-gauge needle (25 g) despite the anticoagulation [20].

## 2. Types of Pancreatic Cystic Neoplasms

### 2.1. Pseudocysts

Pseudocysts are nonneoplastic pancreatic lesions with no malignancy potential, defined as a collection of amylase-rich fluid containing inflammatory cells, blood, and debris. Their walls are made of fibrous and granulation tissue, as they lack a true wall with epithelium lining [1]. The pseudocysts can develop within the pancreatic parenchyma or in the adjacent area. They develop in the pancreatic necrosis area following an attack of acute pancreatitis or in chronic pancreatitis due to obstruction of the pancreatic duct [21]. The exact size and location of the lesion, as well as the communication of the pseudocyst with the pancreatic ductal system, can be visualized using EUS, MRCP (magnetic resonance cholangiopancreatography), or ERCP (endoscopic retrograde cholangiopancreatography) [3]. MRCP is the most accurate and sensitive imaging technique in the context of visualizing the anatomy of the pancreatic duct, its possible connection to the pseudocyst, and the prediction of possible drainage. ERCP, despite being limited by its invasive nature and possible complications during the procedure, is a gold standard in the diagnosis of pancreatic duct disruption. It can also be used for therapeutic purposes. EUS, on the other hand, has the highest sensitivity and specificity (93–100% and 92–98%, respectively) in the distinction between pseudocyst and acute fluid collection [22]. Pseudocysts may be asymptomatic or present with a variety of symptoms, such as pain, early satiety, upper gastrointestinal bleeding, nausea, and vomiting [22,23,24]. Uncomplicated and asymptomatic pseudocysts should be monitored through US without the specific treatment [25]. The development of symptoms or complications, such as infection, hemorrhage, and rupture, require intervention. Clinical evidence shows that these complications significantly increase the risk of severe outcomes, including sepsis from infection, internal bleeding from hemorrhage, and peritonitis from rupture [26,27]. If technically possible, the preferred approach is the endoscopic treatment, as it is more effective, it has a lower reintervention rate and fewer complications, it leads to shorter hospital stays, and it has comparable morbidity and mortality compared to percutaneous drainage [28,29,30]. Percutaneous drainage should be considered as the second-line treatment if endoscopic access is unavailable [31].

### 2.2. Serous Cystic Neoplasm

Serous cystic neoplasms (SCNs), also known as serous cystadenomas, are benign pancreatic tumors with an exceptionally low malignancy potential of 0.1% [32,33]. SCNs represent about 30% of all cystic neoplasms of the pancreas. They occur most commonly in woman in the sixth decade of life [1,34,35]. They are defined as well-bordered masses encased in fibrous capsules that hold many small fluid-filled cysts in the classic “honeycomb” pattern, often arranged around a fibrous stellate scar [35]. Although SCNs can increase in size, the rate of growth is slow, and symptom occurrence is uncommon [11]. There is no surveillance needed in asymptomatic patients, as malignant progression is negligible [7,32]. A follow-up is only required when the diagnosis of the cyst is uncertain. In that case, surveillance should include clinical evaluation, serum CA 19.9 measuring, and MRI and/or EUS imaging every six months for the first year and then yearly [11]. Surgical resection of SCNs is recommended only in patients with symptomatic SCNs caused, for example, by the compression of adjacent organs [11]. Following resection, no further surveillance is needed [7].

### 2.3. Solid Pseudopapillary Epithelial Neoplasm

Solid pseudopapillary epithelial neoplasm (SPEN) of the pancreas is a rare neoplasm that accounts for 1–2% of all exocrine pancreatic tumors. It occurs predominantly in females in the second or third decades of their lives [36,37,38] The majority of SPENs of the pancreas have been characterized as benign, although 10–15% exhibit malignant behavior (e.g., the risk of recurrence) and metastases [35,36,39]. SPENs are usually well-circumscribed, large, solitary, ovoid tumors. The large lesions may show cystic degeneration and/or intertumoral hemorrhage, while smaller lesions may be solid and firm. They may be asymptomatic, smaller lesions and only discovered by chance during imaging tests carried out for unrelated purposes. When symptomatic, patients can experience nausea, vomiting, and nonspecific abdominal pain [36]. The preferred diagnostic tool to differentiate SPEN is MRI with MRCP over CT because of its lack of radiation, higher reported prevalence of detecting PCN (13.5–45% for MRI/MRCP compared to 2.1–2.6% for CT), higher accuracy for identifying specific types of PCNs (40–95% to 40–81%) [11], and higher sensitivity and specificity in distinguishing cancerous and precancerous from benign lesions (93% and 85% for MRI and 62% and 64%, respectively) [40]. Specifically, MRI/MRCP has demonstrated higher sensitivity and specificity for identifying pancreatic lesions, with sensitivity up to 97% and specificity of 90% for detecting SPENs. This higher diagnostic accuracy makes MRI/MRCP more reliable than CT for distinguishing SPENs from other pancreatic cysts and neoplasms [41,42]. However, CT can also be used in patients who cannot undergo MRI [7]. According to international consensus, all SPENs should be surgically resected due to their malignant potential [7,11,43]. EUS-guided biopsy is the diagnostic gold standard in preoperative pathological diagnosis, and to ensure its accuracy, the biopsy should be taken from several sites [44,45]. The prognosis is excellent, with the overall 5-year survival rate being about 97% [46].

### 2.4. Cystic Pancreatic Neuroendocrine Tumors

Cystic pancreatic neuroendocrine tumors (cPNETs) are rare tumors that account for 13–17% of pancreatic neuroendocrine tumors [38,47]. They have the same prevalence in both men and women and may vary in size [47,48]. Cystic PNETs are divided into two subtypes: nonfunctional, which is the more common one, and functional. The functional subtype comprises tumors’ secretions, including insulin, gastrin, glucagon, and somatostatin. Functional cPNETs can present symptoms depending on the particular hormone being secreted [49,50]. If functional cPNET is suspected, a targeted biochemical evaluation based on the type of tumor’s functional status should be included. The biochemical markers include insulin, C-peptide, gastrin, pancreatic polypeptide, vasoactive intestinal polypeptide, glucagon, and calcitonin [51]. Cystic PNETs can be sporadic or associated with multiple endocrine neoplasia type 1 (MEN-1), an autosomal-dominant disorder caused by mutations in the tumor suppressor gene MEN1, characterized by the combined occurrence of tumors of the parathyroid glands, pancreatic islet cells, and the anterior pituitary gland [49,52,53,54,55]. The risk of malignancy for cPNETs is about 20% and smaller than its solid counterpart [11,56]. EUS-guided FNA with cytology is considered the most accurate method to diagnose cPNET, with a specific diagnosis in 71% of cases [1]. Even with low cellularity, the characteristic classic endocrine morphology in cytologic preparation is enough to diagnose an endocrine tumor [57]. The management of cPNET depends on the size of the lesion and its characteristics. The non-functional cysts > 20 mm should undergo surgery, while for non-functional asymptomatic cysts ≤ 20 mm without any malignant behavior, surveillance is recommended [11,50]. The surveillance should comprise EUS, MRI, or CT imaging every 6–12 months. If the change in size is >5 mm or the final diameter is >20 mm, the cyst should be resected. All functional cPNETs should be surgically removed in the absence of metastases [50]. Also, the diagnosis of MEN-1 should be ruled out [49].

### 2.5. Mucinous Cystic Neoplasm

Mucinous cystic neoplasm (MCN) is a rare pancreatic tumor that occurs mostly in women, predominantly in middle age. They are usually found incidentally and are mainly located in the body or tail of the pancreas. MCNs are defined as mucin-producing cysts that are characteristically surrounded by ovarian-type stroma [7,58,59]. In imaging techniques, they present as single spherical masses that can be unilocular or multilocular and contain either thick mucin and/or hemorrhagic–necrotic material mixture. Peripheral calcification may be seen, in contrast to the stellate-type calcification of the SCN [60]. The European guidelines recommend MRI as the imagining modality for MCN, as well as other PCNs, although CT can also be used alone or in conjunction with MRI to identify calcification, tumor staging, or for diagnosing postoperative recurrent disease. They also recommend additionally the combination of CEA, cyst fluid amylase/lipase levels, and cytology to achieve the highest accuracy in distinguishing between mucinous and non-mucinous cysts [11].

MCNs show low rates of malignant transformation in asymptotic lesions without worrisome features in imaging techniques, like mural nodules or eggshell calcification, and they do not surpass a certain size, which varies depending on the guidelines. According to the American Gastroenterological Association (AGA) [61] and the Fukuoka guidelines [62], a size ≥ 3 cm increases the risk of malignancy [61], while The European Study Group Guidelines [11] suggest another cut-off point of 4 cm. Cysts measuring less than those values without other risk factors can be safely surveilled. The surveillance protocols and the detailed management of MCN based on international guidelines are presented in the ”Guidelines” paragraph.

### 2.6. Intraductal Papillary Mucinous Neoplasms (IPMNs)

Intraductal papillary mucinous neoplasms (IPMNs) are the most common PCNs. They are characterized by the production of thick mucin and are frequently located in the head of the pancreas. IMPNs are divided into “main duct type”, which arise from the main pancreatic duct, “branch duct type”, which arise from the pancreatic duct branches, and a combination of both, named “mixed type” [1]. The differentiation between these types is important, as branch duct IPMN (BD-IPMN) has lower malignant potential and a different management protocol compared to other types, which have similar risks of malignant transformation and the same treatment [11]. In up to 40% of cases, IMPNS are multifocal, with the majority of the cysts being genetically unique and potentially progressing independently. Nevertheless, there is no proof that multifocal IPMNs have an increased risk of malignant transformation [7]. IMPNs have moderate to high malignancy potential, with the side branch type having a 22% risk of malignant transformation [63] and the main duct 38–68% [64] IMPN. EUS and cytology results collected through EUS-FNA are crucial for detecting high-grade dysplasia or invasive carcinoma [11,61,65,66,67]. At duodenoscopy, the extrusion of mucin from the Vater ampulla is nearly always indicative of IPMN [1]. The management of IPMN based on international guidelines is presented in the ”Guidelines” paragraph.

A short characterization of different PCLs is presented in Table 1.

## 3. Guidelines

The management of pancreatic cystic lesions (PCLs) is guided by five principal sets of recommendations: the guidelines issued by the American Gastrointestinal Association (AGA), the American College of Gastroenterology (ACG), the American College of Radiology (ACR), UEG, and the International Association of Pancreatology (IAP)/Fukuoka guidelines. These guidelines are based on retrospective studies, which often lack controls for various tumor- and patient-specific factors [62].

### 3.1. Assessment of Pancreatic Cysts

The assessment of a patient with an inadvertent pancreatic cyst should begin with a thorough history, focusing on episodes of acute or chronic pancreatitis. Additionally, the patient’s family history is crucial, particularly for instances of pancreatic cancer or hereditary cancer syndromes linked to an elevated risk of pancreatic malignancy. This targeted approach gathers pertinent information essential for guiding further diagnostic and management decisions in the context of incidental pancreatic cysts [61].

### 3.2. Management of Intraductal Papillary Mucinous Neoplasms (IPMNs)

The International Consensus Guidelines for the Management of Intraductal Papillary Mucinous Neoplasms (IPMNs) were initially introduced in 2006 at the 11th International Association of Pancreatology (IAP) meeting in Sendai, Japan. These “Sendai Guidelines” recommended the resection of main duct IPMNs (MD-IPMNs) and mucinous cystic neoplasms (MCNs) in all surgically fit patients. The management of branch duct IPMNs (BD-IPMNs) posed challenges due to a high rate of benign cases. The 2012 “Fukuoka Guidelines” revised surgical indications and follow-up methods for BD-IPMNs and introduced molecular markers for better differentiation. It also included “worrisome features” (WFs) and “high-risk stigmata” (HRS), which correspond to Europe’s “relative indications for surgery” and “absolute indications for surgery”, respectively [62,69] The 2016 “Revised Fukuoka Guidelines” further refined the definition of “malignant” IPMN to include high-grade dysplasia and invasive carcinoma, emphasizing the presence of mural nodules as a strong predictor of malignancy, with a specified cut-off size of 5 mm. Additional factors, such as enhancing mural nodules, elevated CA 19-9 levels, and rapid cyst growth, were incorporated as WFs, while obstructive jaundice in a patient with a cystic lesion of the pancreatic head, enhanced mural nodule > 5 mm, and an MPD size of >10 mm were considered HRS. The revised guidelines advocate for a more conservative approach in BD-IPMN management but long-term surveillance due to the high incidence of concomitant pancreatic ductal adenocarcinoma (PDAC).

The latest guidelines, published in 2024 following the 2022 meeting in Kyoto, Japan, included updates on HRS and WFs, surveillance protocols for non-resected IPMN, post-resection surveillance, and the investigation of molecular markers in cyst fluid [65,70,71,72,73]. The novelties are the inclusion of new onset or acute exacerbation of diabetes mellitus (DM) and a cystic growth rate ≥ 2.5 mm/year as WFs [65,70,71].

According to the Kyoto 2024 guidelines [65], IPMN surgical resection should be considered if any of the HRS, which include dilation of MPD ≥ 10 mm, a contrast-enhancing mural nodule ≥ 5 mm, or a solid component in MRI, suspicious or positive results of cytology, or obstructive jaundice caused by a cyst of the head of the pancreas, are present. The resection should be also considered in patients with multiple WFs, patients with at least one WF who are young and fit for surgery, and patients with repeated acute pancreatitis. The WFs are the following: 1. acute pancreatitis; 2. increased serum level of CA 19-9; 3. new onset or acute exacerbation of diabetes within the past year; 4. cyst size ≥ 30 mm; 5. contrast-enhancing mural nodule ≥ 5 mm; 6. thickened or contrast-enhancing cyst walls; 7. MPD dilation ≥ 5 mm and <10 mm; 8. abrupt change in caliber of pancreatic duct with distal pancreatic atrophy; 9. lymphadenopathy; and 10. cystic growth rate ≥ 2.5 mm/year. The indications for surgery based on the “International evidence-based Kyoto guidelines for the management of intraductal papillary mucinous neoplasm of the pancreas” [65] are summarized in Table 2.

Post-resection surveillance is recommended yearly for patients without additional risk factors and every six months for those with high-grade dysplasia (HGD) or a family history of pancreatic cancer. This concerns patients suitable for additional therapeutic intervention. Surveillance methods include imaging studies (MRI, MDCT, EUS), physical examination, and blood examination, including tumor markers (CA 19-9 and CEA) and hemoglobin A1c (HbA1c), which is similar to the surveillance of non-resected IPMN [74].

For non-resected BD-IPMNs, the surveillance recommendations vary based on the cyst size. Following the initial 6-month follow-up, a surveillance interval is recommended as follows: 18 months for lesions measuring < 20 mm, 12 months for lesions measuring > 20 mm and <30 mm, and 6 months for lesions ≥ 30 mm. MRI along with physical examination, HbA1c serum level, and assessment of CA 19.9 are the preferred methods, although other methods, like multi-detector computer tomography, MCTD, and EUS, can also be used. Surveillance may be ceased for patients with small BD-IPMNs < 20 mm that remain stable and have no WFs after 5 years of surveillance. The patient’s health, fitness for surgery, and life expectancy must be considered in surveillance continuation. Young patients and those with genetic risks or family history of pancreatic cancer should be further surveilled.

The European evidence-based guidelines on pancreatic cystic neoplasms (2018) [11] present evidence-based guidelines for the management of PCN, addressing a notable gap in the existing literature. A collaborative effort by various European gastrointestinal and pancreatic associations involving the European Study Group on Cystic Tumours of the Pancreas, United European Gastroenterology, the European Pancreatic Club, the European-African Hepato-Pancreato-Biliary Association, European Digestive Surgery, and the European Society of Gastrointestinal Endoscopy resulted in comprehensive guidelines covering biomarkers, radiology, endoscopy, IPMN, MCN, serous cystic neoplasm, rare cysts, neoadjuvant treatment, and pathology.

For radiologically suspected IPMN, the European guidelines [11] recommend surgery for all surgery-fit patients with at least one absolute indication for surgery, including positive cytology for malignant/high-grade dysplasia, a contrast-enhancing mural nodule in MRI ≥ 5 mm, a dilation of MPD ≥ 10 mm, and jaundice caused by the cyst. Surgery is also recommended for patients with at least two relative indications for surgery and for patients without any significant co-morbidities who have at least one relative indication. Relative indications for surgery in IPMN include an MPD diameter between 5 and 9.9 mm, a cyst diameter ≥ 40 mm, a growth rate ≥ 5 mm/year, a serum CA 19.9 > 37 U/mL, new-onset DM, a contrast-enhancing mural nodule < 5 mm, and whether the cause is cyst acute pancreatitis. After surgery, lifelong follow-up is recommended in surgically fit patients. The follow-up should consist of clinical evaluation, serum CA 19.9 measuring, and MRI and/or EUS imaging. Post-resection surveillance for patients with resected IPMNs with high-grade dysplasia or MD-IPMNs should be executed every 6 months for the first 2 years and then yearly. Patients post-resection, as well as patients with no non-resected IPMNs with no absolute or relative indications for surgery, should be surveilled every 6 months for the first year and then yearly. Patients with non-resected IPMN with only one relative indication for surgery with a short life expectancy or significant co-morbidities should be intensively surveilled every 6 months. Patients unfit for surgery do not require any surveillance [11]. The indications for IPMN surgery based on the European evidence-based guidelines on pancreatic cystic neoplasms are presented in Figure 1.

The choice of surgical procedure is based on preoperative imaging features (presence of HRS) and intraoperative frozen section findings. When invasive carcinoma is suspected, radical pancreatectomy with lymph node dissection should be performed. When a non-invasive lesion is suspected, middle pancreatectomy or spleen-preserving pancreatectomy should be performed. Other approaches, such as laparoscopic or robotic pancreatectomy, can be also considered based on local centre experience. Indications for radical or organ-preserving treatment are the same in BD-IPMN, MD-IPMN, and mixed IPMN [65].

### 3.3. Management of Mucinous Cystic Neoplasms (MCNs)

The 2015 “American Gastroenterological Association Institute Guideline on the Diagnosis and Management of Asymptomatic Neoplastic Pancreatic Cysts” [61] does not specifically address the management of MCNs but provides recommendations for all asymptomatic pancreatic cysts. Those guidelines [61] suggest MRI surveillance of cysts smaller than 30 mm without a solid component or a dilated pancreatic duct. If the size or characteristics of the cyst do not change, surveillance should consist of one examination in the first year and every 2 years for a total of 5 years. Pancreatic cysts with at least 2 high-risk features, including a solid component, a dilated main pancreatic duct, or a size greater than 30 mm, should be examined with EUS-FNA. Patients examined through EUS-FNA without any disquieting findings should undergo surveillance after 1 year and then every 2 years. For patients unfit for surgery and those with cysts with no significant changes after 5 years of surveillance, surveillance should be discontinued. For patients with both a solid component and a dilated pancreatic duct and/or concerning findings upon EUS-FNA, surgical resection should be performed. After surgical resection, cysts without high dysplasia or malignancy do not require further surveillance, while those with dysplasia or invasive cancer should undergo MRI every 2 years.

The management of MCN is also addressed in the 2018 “ACG Clinical Guideline: Diagnosis and Management of Pancreatic Cysts” [7]. For presumed MCNs, the ACG recommends surveillance for asymptomatic cysts, preferably using MRCP, although in patients who cannot or choose not to have MRI, EUS can also be used. The duration and frequency of surveillance are not stated in those guidelines. During surveillance, if the patient experiences new onset or worsening of DM, or a lesion increases in size > 3 mm/year, the patient should undergo immediate MRI or EUS with or without FNA. EUS/FNA should also be performed in patients with symptoms caused by a cyst (jaundice and acute pancreatitis), significantly elevated CA 19-9 serum, worrisome imaging findings (mural nodule or solid component, dilation of main pancreatic duct of >5 mm, and lesions ≥ 30 mm), or the presence of either high-grade dysplasia or pancreatic cancer upon cytology. The guidelines do not provide indications for surgery, instead preferring referral to a multidisciplinary group in specialized centers for further management.

The European evidence-based guidelines (2018) [11] also address the management of MCNs. The guidelines recommend surveillance for asymptomatic MCN < 40 mm without a contrast-enhancing mural nodule in MRI. Surveillance should consist of MRI and/or EUS every 6 months for the first year and then annually for the rest of the patient’s life if they are fit for surgery. MCN ≥ 40 mm, symptomatic MCN, and MCN with malignant risk factors, like a contrast-enhancing mural nodule, should always be resected.

The comparison between the different guidelines for MCN, including surveillance, indications for surgery, and follow-up, is presented in Table 3.

### 3.4. Pancreatic Cyst Fluid (PCF) Analysis

While fluid cytology has traditionally been employed for diagnosing PCLs, its definitive diagnostic yield remains limited, particularly in non-malignant mucinous cysts. PCF analysis includes CEA, which is commonly utilized as a prominent biomarker. However, CEA has limitations; while it offers high specificity (up to 96% at levels above 192 ng/mL), its sensitivity is relatively low, between 58% and 63% [14]. A pivotal study revealed that a lower cut-off level of cyst fluid CEA, specifically 45 ng/mL, markedly improves the diagnostic precision for identifying mucinous PCLs, with a noted sensitivity and specificity exceeding 88% and 96%, respectively. It is important to note that while elevated CEA levels (≥45 ng/mL) are strongly associated with mucinous cysts, they do not specifically indicate malignancy. Instead, they help differentiate between mucinous (potentially pre-malignant or malignant) and non-mucinous (generally benign) cystic lesions. The validated CEA cut-off level of 30.7 ng/mL suggests the presence of IPMNs, though it is not exclusively diagnostic [76]. Therefore, a CEA level at or above 45 ng/mL strongly suggests a mucinous cyst but does not exclude IPMNs, as both IPMNs and MCNs can exhibit such levels. The range of CEA levels from 30.7 ng/mL to 45 ng/mL may include IPMNs as well as other mucinous cysts, but as the CEA level approaches 45 ng/mL, the likelihood of a mucinous lesion increases [2]. It is important to note that while these CEA thresholds aid in initial identification, further diagnostic methods are necessary to accurately categorize the type of mucinous cyst [75,77].

In 153 pancreatic cyst fluid samples, it was demonstrated that mucinous cysts have significantly lower glucose levels compared to nonmucinous ones, with median values of 19 mg/dL versus 96 mg/dL. Using a glucose threshold of ≤50 mg/dL, the sensitivity, specificity, and accuracy for distinguishing mucinous cysts were 92%, 87%, and 90%, respectively. In the same study, CEA had a sensitivity of 58%, a specificity of 96%, and an accuracy of 69% [78]. Therefore, recent guidelines recommend the measurement of glucose levels in pancreatic cyst fluid for the routine diagnosis of pancreatic mucinous cysts.

In pancreatic cyst fluid, the most widely used genetic tests are KRAS and GNAS mutations with advanced techniques, like next-generation sequencing (NGS). KRAS mutations are a hallmark of IPMNs found in about 68% of these lesions and 78% of IPMNs with adenocarcinoma [79]. These mutations offer high specificity (98%) for mucinous differentiation, helping to distinguish mucinous cysts from other types, thus aiding in assessing their malignant potential [4,79]. GNAS mutations are also prevalent in IPMNs, detected in 39% of cases and 22% of IPMNs with adenocarcinoma. The presence of both KRAS and GNAS mutations increases diagnostic accuracy, enhancing the differentiation between benign and potentially malignant cysts [1,79,80,81].

Tumor markers play a crucial role in the diagnosis and management of PCLs. They provide valuable information that complements imaging findings and helps refine the diagnosis, prognosis, and treatment strategy.

Vascular Endothelial Growth Factor (VEGF) levels can indicate the presence of serous cystadenomas (SCAs), distinguishing them from other cyst types, which is critical because SCAs are generally benign and managed conservatively unless symptomatic [80]. Mucin (MUC) genes, such as MUC1 and MUC2, are associated with different types of pancreatic cysts, with MUC2 typically expressed in IPMNs and MUC1 associated with more aggressive cystic neoplasms [80,81]. The integration of these tumor markers with advanced imaging and molecular techniques enhances diagnostic accuracy and helps in making informed decisions regarding the management of pancreatic cystic lesions, ultimately improving patient outcomes.

A 2021 study [79] demonstrated that testing for KRAS and GNAS mutations achieved a sensitivity of 94.7% and a specificity of 100% in differentiating mucinous pancreatic cystic lesions. This highlights its effectiveness as a standalone diagnostic test. Additionally, GNAS mutations are found exclusively in branch duct IPMNs, making GNAS a valuable marker for further classification after identifying mucinous lesions [4]. The combined analysis of KRAS and GNAS mutations significantly enhances the diagnostic accuracy for BD-IPMN. A large meta-analysis involving 785 BD-IPMN lesions reported a sensitivity of 94%, a specificity of 91%, and an overall diagnostic accuracy of 97% [82]. Additionally, a separate study highlighted that the combination of KRAS and GNAS mutations achieved a specificity of 98% and a sensitivity of 84% in diagnosing BD-IPMN [83]. Furthermore, a 2016 study demonstrated that KRAS and/or GNAS mutations were identified with 100% specificity and sensitivity for diagnosing BD-IPMNs [84]. Some studies have explored the role of additional mutations, such as BRAF. Ren et al. (2021) found that in KRAS-negative BD-IPMNs, BRAF mutations with concurrent GNAS mutations were found, indicating an alternative activation mechanism in the Ras pathway [85].

Other genetic mutations evaluated in PCF are TP53, and CDKN2A, commonly associated with high-grade dysplasia and invasive cancer in pancreatic cysts. TP53 mutations appear in 44% of pancreatic ductal adenocarcinomas (PDACs), while CDKN2A mutations are found in 35%. These genetic alterations are crucial for identifying high-risk lesions that require more aggressive treatment or closer monitoring [2,76]. Additionally, studies have shown that methylated DNA markers in cyst fluid can accurately detect precancerous and early cancerous lesions [82]. For example, Hong et al. (2008) [84] discovered increasing levels of improper DNA methylation in genes, such as SPARC and TSLC1, correlated with the progression of pancreatic neoplasms. More recently, Hata et al. (2022) [85] identified that hypermethylation of the SOX17 gene could accurately detect advanced neoplasia in branch duct IPMNs, achieving a sensitivity and specificity of 83% and 81.8%, respectively. Majumder et al. [86] also validated a panel of methylated DNA targets, such as TBX15 and BMP3, which could distinguish between high-grade and low-grade lesions with high accuracy. These findings underscore the potential of DNA methylation as a molecular tool for early detection and risk stratification in pancreatic cancer.

Detecting KRAS, GNAS, TP53, and CDKN2A mutations in cyst fluid enhances the ability to differentiate between benign and malignant cysts. The high specificity of KRAS and GNAS mutations for mucinous cysts and IPMNs has been well-documented, with sensitivity reaching 89% and specificity up to 100%. Additionally, these mutations have shown high specificity (>90%) for mucinous cysts and IPMNs, although sensitivity is relatively lower at 65% [87]. The combined use of CEA levels and KRAS mutations improves the sensitivity for detecting mucinous cysts to 83%, albeit with a slight reduction in specificity to 85% [79]. Preliminary studies of various DNA mutation panels have suggested high sensitivity and specificity for identifying cysts with high-grade dysplasia or malignancy. This indicates that incorporating DNA mutation analysis into the diagnostic process can provide a more accurate stratification of pancreatic cystic lesions, aiding in the differentiation between benign and malignant cysts [88].

The combination of these genetic markers enhances the diagnostic capabilities of EUS-FNA, allowing for more accurate stratification of cystic lesions and personalized treatment plans. Detecting KRAS and GNAS mutations can prevent unnecessary surgeries for benign cysts [83]. Studies have shown that KRAS mutations have a specificity of 100% but a sensitivity of 54% for mucinous differentiation. When combined with CEA analysis, the sensitivity increases to 83%, while maintaining a high specificity of 85% [79].

Conversely, identifying these mutations in high-risk mucinous cysts ensures timely surgical intervention, thereby preventing the progression to invasive cancer and improving patient outcomes [79,89]. For instance, a patient with detected KRAS and GNAS mutations may undergo early resection of an IPMN, averting potential malignancy and aligning with personalized treatment strategies [89,90].

High levels of amylase, generally greater than 250 U/L, suggest a pseudocyst rather than a neoplastic lesion. These markers aid in differentiating mucinous from non-mucinous cysts and in identifying pseudocysts associated with pancreatitis [5,68].

A summarized analysis of pancreatic cyst fluid of cystic pancreatic lesions is included in Table 4.

## 4. Advancements in EUS

There have been significant advancements in EUS techniques in recent years.

Contrast-enhanced harmonic EUS utilizes ultrasound contrast agents that enhance the backscatter of ultrasound signals from blood, providing better visualization of vascular patterns within lesions. Studies indicate that CH-EUS can effectively distinguish between different types of pancreatic cystic lesions, such as serous cystadenomas, mucinous cystic neoplasms, and intraductal papillary mucinous neoplasms, based on their vascularity patterns [93].

The incorporation of contrast-enhanced harmonic EUS (CH-EUS) as a supplementary technique to EUS proves valuable in the identification and characterization of MN. Compared to conventional EUS, CH-EUS increases the sensitivity and specificity of identifying malignant cysts. Recent investigations indicate that CH-EUS exhibits a sensitivity ranging from 60% to 100% and a specificity ranging from 75% to 92.9% in the diagnosis of malignant cysts [54]. The same study showed that when a mural nodule indicates malignancy, CH-EUS significantly outperforms conventional EUS, with a specificity of 75% versus 40% and an accuracy of 84% versus 64% [54]. Additionally, in evaluating the diagnostic performance of CH-EUS versus conventional EUS, a higher AUROC for CH-EUS (0.93) compared to conventional EUS (0.84) indicates that CH-EUS provides better differentiation between malignant and non-malignant cysts [54]. These findings affirm that CH-EUS enhances the detection of pathological alterations in the microvascular architecture, improving the identification of smaller malignant areas and tumoral neoangiogenesis, making it superior to conventional EUS in accurately diagnosing malignant cysts [54]. The guidelines emphasize the potential of CH-EUS in detecting mural nodules, attributing to it a Grade 2C recommendation with strong agreement [94,95]. CH-EUS emerges as a valuable tool for the further evaluation of mural nodules and assessing cyst vascularity, with its ability to highlight features suggestive of malignancy, thus potentially altering management strategies.

Additionally, intraductal ultrasonography emerges as a particularly effective modality for the detection of MN and IPMN, further enhancing diagnostic capabilities in this clinical context [96]. Intraductal ultrasonography (IDUS) is an advanced endoscopic ultrasound technique that provides high-resolution imaging of the pancreatic and biliary ducts. The procedure involves inserting a high-frequency ultrasound probe (12–30 MHz) through an endoscope into the ductal system, allowing for detailed visualization of the ductal walls and surrounding tissues. IDUS is particularly effective for detecting small lesions, assessing ductal invasion by tumors, and distinguishing between benign and malignant strictures. Compared to standard EUS, IDUS offers superior resolution, making it a valuable tool in the comprehensive evaluation of pancreatic and biliary diseases [97].

EUS is crucial in differentiating pancreatic cysts from other pathologies, such as gastric duplication cysts (GDCs), which are rare and challenging to diagnose with standard ultrasound. GDCs are uncommon congenital anomalies, often presenting with non-specific symptoms like abdominal pain, nausea, and weight loss, complicating their diagnosis.

Advancements in ultrasonography and EUS have significantly improved the diagnostic accuracy for rare intestinal duplications, including GDCs. EUS provides high-resolution, real-time imaging that surpasses traditional CT scans, offering detailed visualization of cyst walls and internal structures characteristic of GDCs. This precision helps exclude other diagnoses, like gastrointestinal stromal tumors (GISTs) or malignancies [98,99,100].

EUS-guided fine-needle aspiration (FNA) further enhances diagnosis by allowing cytological analysis and fluid testing, confirming benign cysts and ruling out malignancy. In cases reported by Mohamad Abdalkader et al. and Seijo et al., EUS demonstrated high diagnostic accuracy and guided effective surgical decisions, preventing unnecessary procedures and aiding precise treatment planning [98,100].

Thus, EUS not only improves diagnostic accuracy but also significantly impacts patient management by providing essential information for appropriate therapeutic interventions, especially for rare pathologies like GDCs, leading to timely and effective treatments and reducing complication risks [40,101,102].

Confocal laser endomicroscopy offers a significant advantage in the evaluation of PCNs by providing high-resolution, real-time microscopic imaging of cyst walls. This technique allows for the detailed visualization of cellular structures and tissue architecture, aiding in the differentiation between benign and malignant cysts. Studies have shown that nCLE can effectively identify characteristic features of malignancy, such as papillary projections and epithelial bands, particularly in intraductal papillary mucinous neoplasms and mucinous cystic neoplasms [103,104,105].

Research has demonstrated the utility of nCLE in guiding biopsies by pinpointing the most suspicious areas within a cyst, thereby improving the accuracy of the diagnosis and subsequent management decisions. The safety profile of nCLE is also well-established, with minimal complications reported, making it a viable option for routine clinical use [103,105]

Recent studies highlight the integration of next-generation sequencing and advanced imaging techniques like EUS-guided confocal laser endomicroscopy, which may significantly enhance the precision of diagnosing pancreatic cysts and predicting their malignant potential [106]. For instance, McCarty et al.’s systematic review revealed that molecular analysis of cyst fluid for KRAS and GNAS mutations, combined with nCLE imaging, improves diagnostic accuracy, reduces unnecessary surgeries, and ensures prompt treatment of high-risk lesions [79,88,107]. A systematic review and meta-analysis by Napoleon et al. evaluated the diagnostic accuracy of EUS-nCLE and found that while it improves the visualization of cystic lesions, its standalone diagnostic utility remains limited without molecular analysis [108]. It is suggested to combine the high-resolution imaging capabilities of nCLE and the genetic insights provided by NGS. Studies by Al-Haddad et al. and Law et al. have demonstrated that integrating molecular markers with advanced imaging techniques can significantly reduce unnecessary surgeries by accurately identifying high-risk lesions that warrant intervention [88]. These advancements suggest a future where management strategies for PCNs are increasingly personalized, offering targeted treatments based on precise molecular and imaging data [2].

EUS-guided elastography is a pivotal technique for the evaluation of pancreatic cysts, particularly when traditional imaging methods are inconclusive. This technology includes strain elastography, which measures tissue deformation under applied pressure, and shear wave elastography, which evaluates the speed of shear waves passing through the tissue. Studies have demonstrated that strain elastography has a sensitivity of 91% and a specificity of 85% for detecting malignant pancreatic lesions, while shear wave elastography offers a sensitivity of up to 92% and specificity up to 89% [109,110]. These high accuracy rates make elastography essential for differentiating between benign and malignant lesions, guiding biopsy decisions, and improving diagnostic precision. It is particularly useful in cases with ambiguous imaging results or high-risk features, such as large cyst size, mural nodules, or main pancreatic duct dilation [111,112]. The integration of elastography with EUS enables more targeted biopsies and personalized management strategies, reducing unnecessary surgeries and ensuring timely interventions for high-risk patients.

In conclusion, the management of PCLs requires accurate identification of cyst types to determine their malignancy potential. Mucinous cysts like mucinous cystic neoplasms and intraductal papillary mucinous neoplasms carry a higher risk of becoming malignant and need closer surveillance or intervention. In contrast, serous cystadenomas are mostly benign and require less aggressive management [111,113].

Therefore, a multidisciplinary approach that incorporates these advanced diagnostic technologies and molecular diagnostics is crucial for navigating the complexities of PCL management, ensuring tailored and effective patient care. This approach not only improves diagnostic and prognostic accuracy but also aids in personalizing management strategies to enhance patient outcomes [113].

## 5. Advantages and Disadvantages of ERCP in Pancreatic Duct Management for Patients with Pancreatic Cysts

Endoscopic retrograde cholangiopancreatography (ERCP) is a valuable tool in diagnosing and managing pancreatic duct disorders, including pancreatic cysts. According to Boicean et al.’s [114] study, the procedure offers significant advantages, such as high-resolution imaging for precise diagnosis and therapeutic interventions like stent placement and ductal dilation, which can alleviate symptoms and address underlying causes of pancreatic cysts [115]. As a minimally invasive procedure, ERCP is often preferred over surgical options, reducing recovery time and overall patient morbidity.

However, ERCP also has notable disadvantages, particularly the risk of post-ERCP pancreatitis (PEP), which occurred in 35.8% of patients in the referenced study. Certain factors, such as female sex, Sphincter of Oddi dysfunction, and complex procedures, increase this risk [116]. Additionally, ERCP can cause mechanical, thermal, and chemical trauma to the biliary and pancreatic ducts, leading to complications like papillary edema, which obstructs pancreatic fluid outflow and triggers pancreatitis [117]. Other risks include infections and bleeding, particularly when interventions like sphincterotomy are performed [118].

Thus, while ERCP is an effective diagnostic and therapeutic tool for pancreatic cyst management, the potential risks, especially PEP, must be carefully weighed against the benefits for each patient.

## 6. Conclusions

The management of pancreatic cystic lesions (PCLs) necessitates precise identification to determine malignancy potential and appropriate intervention. This review underscores advancements in diagnostic techniques and guidelines that enhance the differentiation, assessment, and treatment of PCLs.

EUS and EUS-FNA are vital for evaluating pancreatic cysts with high sensitivity and specificity. Techniques like nCLE and EUS-guided elastography further improve diagnostic accuracy. Advanced genetic testing, including KRAS and GNAS mutations, enhances diagnostic precision and risk stratification.

Management guidelines emphasize a tailored approach based on cyst characteristics and patient factors. High-risk lesions warrant surgical intervention, while asymptomatic, low-risk cysts are managed with surveillance. The integration of molecular diagnostics and advanced imaging reflects a shift toward precision medicine.

A multidisciplinary approach combining advanced diagnostics, molecular testing, and evidence-based guidelines is essential for effectively managing PCLs. This strategy improves diagnostic accuracy, personalizes management, and enhances patient outcomes. Ongoing research and technological advancements will continue to refine the diagnosis and treatment of PCLs.

## Figures and Tables

**Figure 1 jcm-13-04644-f001:**
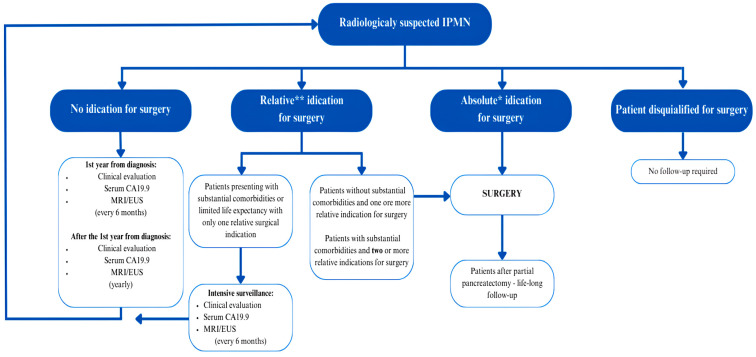
Management of IPMN—indications for surgery—European Study Group guidelines [11]. * Absolute indication for surgery: positive cytology for malignancy/HGD, solid mass, jaundice (tumor-related), enhancing mural nodule (≥5 mm), MPD dilatation ≥ 10 mm. ** Relative indications for surgery: growth rate ≥ 5 mm/year, increased levels of serum CA 19.9 (>37 U/mL), MPD dilatation 5 to 9.9 mm, cyst diameter ≥ 40 mm, new onset of diabetes mellitus, acute pancreatitis (caused by IPMN), enhancing mural nodule (<5 mm). IPMN—intraductal papillary mucinous neoplasm; EUS—endoscopic ultrasound.

**Table 1 jcm-13-04644-t001:** Characterization of different PCLs [1,62,68].

Type of PCL	Age	Sex	Location in Pancreas	Malignancy Potential	Connection to Main Duct	Characteristics Upon Imaging
Pseudocysts	Any	Equal	Anywhere	None	Some	Well-circumscribed, oval or round, anechoic upon EUS
SCN	40–60	75% F	Tail or body	Low	No	Honeycomb; some have central scar
SPEN	20–30	90% F	Anywhere	Moderate to high	No	Well-demarcated, mixed solid–cystic tumors
MCN	40–50	F	Tail	High	No	May be unilocular or septated; some have peripheral calcifications
IPMN	60–70	Equal	Mainly head	Moderate to high	Yes	Dilatation of PD

PCL—pancreatic cystic lesion; SCN—serous cystic neoplasm; SPEN—solid pseudopapillary epithelial neoplasm; MCN—mucinous cystic neoplasm; IPMN—intraductal papillary mucinous neoplasm; F—females; EUS—endoscopic ultrasonography.

**Table 2 jcm-13-04644-t002:** Indications for surgery of IPMN based on the “International evidence-based Kyoto guidelines for the management of intraductal papillary mucinous neoplasm of the pancreas” [65].

1 ≥ HRS	2 ≥ WFs or 1 ≥ WF in Young Fit-for-Surgery Patients or Repeating Acute Pancreatitis
Dilation of MPD ≥ 10 mm. Contrast-enhancing mural nodule ≥ 5 mm or solid component in MRI. Suspicious or positive results of cytology (if performed). Obstructive jaundice caused by cyst of the head of the pancreas.	Acute pancreatitis. Increased serum level of CA 19-9. New onset or acute exacerbation of diabetes within the past year ^1^. Cyst size ≥ 30 mm. Contrast-enhancing mural nodule ≥ 5 mm. Thickened or contrast-enhancing cyst walls. MPD dilation ≥ 5 mm and <10 mm. Abrupt change in caliber of pancreatic duct with distal pancreatic atrophy. Lymphadenopathy. Cystic growth rate ≥ 2.5 mm/year.

^1^ The definitions of “new onset” and “acute exacerbation” have not yet been defined. IPMN—intraductal papillary mucinous neoplasm; HRS—high-risk stigmata; WF—worrisome feature; MPD—main pancreatic duct; MRI—magnetic resonance imaging; CA 19-9—cancer antigen 19-9.

**Table 3 jcm-13-04644-t003:** Management of MCN.

Type of Action	European Guidelines (2018) [11]	ACG Guidelines (2018) [7]	AGA Guidelines (2015) ^1^ [75]
Surveillance	MCN < 40 mm without risk factors and symptoms can be safely surveilled with MRI, EUS, or a combination of both every 6 months for the first year and then annually as long as they are fit for surgery.	Surveillance of surgically fit candidates with asymptomatic cysts. Patients with new-onset or worsening DM, or increase in cyst size > 3 mm/year, should undergo a short-interval MRI or EUS ± FNA.	MRI surveillance during 1st year and then every 2 years for a total of 5 years for cysts < 30 mm without solid component or dilated pancreatic duct and for cysts without concerning EUS-FNA results.
Indication for resection/referral to a multidisciplinary group ^3^	MCN ≥ 40 mm, symptomatic MCN, and MCN with high risk factors, like a mural nodule, regardless of its size.	MCN > 30 mm; MCN with mural nodule or solid component; dilated pancreatic duct > 5 mm; jaundice or acute pancreatitis secondary to the cyst; significantly elevated serum CA 19-9; the presence of HGD or pancreatic cancer upon cytology.	MCN with both a solid component and a dilated pancreatic duct and/or concerning features on EUS and FNA ^2^.
Post-surgery surveillance	No data.	No surveillance for resected MCNs without pancreatic cancer.	No routine surveillance for cysts without HGD or malignancy at resection.

^1^ AGA guidelines pertain only to asymptotic pancreatic neoplastic cysts. ^2^ The indications for EUS-FNA are 2 high-risk features (size ≥ 30 mm, dilated MPD, or the presence of an associated solid component) or significant changes in the properties of the cyst (development of a solid component, increasing size of the pancreatic duct, and/or diameter ≥ 30 mm). ^3^ The ACG prefers referring patients to a multidisciplinary pancreatic group for further evaluation rather than providing indications for surgery. MCN—mucinous cystic neoplasm; ACG—the American College of Gastroenterology; AGA—The American Gastroenterological Association; MRI—magnetic resonance imaging; EUS—endoscopic ultrasonography; DM—diabetes mellitus; FNA—fine-needle aspiration; CA 19-9—cancer antigen 19-9; HGD—high-grade dysplasia; MPD—main pancreatic duct.

**Table 4 jcm-13-04644-t004:** Pancreatic cyst fluid analysis of cystic pancreatic lesions [91,92].

Type of PCL	CEA	Amylase	CA 19-9	KRAS	GNAS
Pseudocysts	Low	High	High	−	−
SCN	Low	Low	Variable	−	−
MCN	High	Low	Variable	+	−
IPMN	High	High	Variable	+	+

PCLs—pancreatic cystic lesions; SCN—serous cystic neoplasm; MCN—mucinous cystic neoplasm; IPMN—intraductal papillary mucinous neoplasm; CA—carbohydrate antigen; CEA—carcinoembryonic antigen.

## Data Availability

Not applicable.

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
