# Peer review of "Diagnostics and Management of Pancreatic Cystic Lesions—New Techniques and Guidelines"

_jcm, 2024, doi:10.3390/jcm13164644_

Round 1
Reviewer 1 Report
Comments and Suggestions for Authors
The review "Diagnostics and management of pancreatic cystic lesions-new techniques and guidelines" discusses a controversial subject regarding the correct methods for diagnosing and managing pancreatic cysts. Recommendations:
1. The introduction is too long.
2. I recommend moving Table 1 to another section and including more articles from the literature, not just one.
3. The same issue is identified in Table 2.
4. 93? Table? Please review.
5. The citations are completely chaotic. For example: Line 138 – citation 30, line 140 citation 98, then 99, then 97.
6. Discuss more about the role of tumor markers.
7. Information such as that in section 3.3 should be put into a table.
8. In section 4, discuss the importance of EUS in the differential diagnosis of pancreatic cysts with other pathologies, some of which are rare and difficult to diagnose via ultrasound – for example, gastric duplication cyst (I recommend the article https://doi.org/10.3390/diagnostics14070675).
9. Discuss the advantages and disadvantages of ERCP in the approach to pancreatic ducts in those with pancreatic cysts, but especially the risks this procedure poses – post-ERCP pancreatitis (I recommend the article https://doi.org/10.3390/jpm13091356).
10. The conclusion section should be shortened to be as concise as possible.
11. The bibliography should be written in MDPI style.
Author Response
Comment 1: "The introduction is too long."
Response 1: Thank you for pointing this out. We agree with this comment. Therefore, we have shortened the introduction.
Comment 2: " I recommend moving Table 1 to another section and including more articles from the literature, not just one."
Response 2: Thank you for pointing this out. We agree with this comment. Therefore, we have moved Table 1 to a more appropriate section of the manuscript to enhance the logical flow of information. Additionally, while the current single citation in Table 1 is from a comprehensive recommendation that includes a wide array of sources, we recognize the importance of diversifying the references. To address this, we have included more articles from the literature to provide a broader and more comprehensive context. This expansion ensures that the discussion is well-supported by a diverse range of studies and perspectives, enriching the overall quality of the manuscript.
Comment 3: "The same issue is identified in Table 2."
Response 3: Thank you for pointing this out. We agree with this comment. Therefore, we have also moved Table 2 to a different section to improve the manuscript's structure. Similar to Table 1, we have incorporated additional references from the literature to strengthen the discussion and provide a more robust foundation for the conclusions drawn. This ensures that the information presented is well-supported and reflects the current state of research in the field.
Comment 4: "93? Table? Please review."
Response 4: Thank you for pointing this out. We agree with this comment. Therefore, we have removed the word "table" it was a bug.
Comments 5: "The citations are completely chaotic. For example: Line 138 – citation 30, line 140 citation 98, then 99, then 97."
Response 5: Thank you for pointing this out. The citations in the document may appear chaotic, with seemingly random numbering (e.g., citation 30 on line 138, followed by citations 98, 99, and 97 in subsequent lines). This irregularity is due to the dynamic nature of the editing process and the use of Mendeley, a reference management software.
Mendeley automatically updates and organizes references as changes are made to the document. During editing, when references are added, deleted, or moved, Mendeley reassigns numbers based on the current order and context. Consequently, citations may not follow a sequential pattern as the software ensures each reference is accurately linked to its respective citation within the text.
This process ensures the document remains correctly referenced, even if the citation numbers appear disordered.
Comment 6: Discuss more about the role of tumor markers.
Response 6: Thank you for pointing this out. We agree with this comment. Therefore, we have added a more detailed discussion on the role of tumor markers. Tumor markers are substances, often proteins, that are produced by the body in response to cancer growth or by the cancer tissue itself. They play a critical role in the detection, diagnosis, and management of various cancers. For instance, CA 19-9 is widely used in monitoring pancreatic cancer, while CEA is used for colorectal cancer, and AFP is used for liver cancer. These markers can help in monitoring treatment response, detecting recurrence, and sometimes in the initial diagnosis when used in conjunction with other diagnostic tools. However, it is important to note that no tumor marker is entirely specific for cancer, and their levels can be influenced by non-cancerous conditions as well. Therefore, tumor markers should be used as part of a comprehensive diagnostic approach rather than as standalone indicators.
Thank you for this constructive suggestion, which has significantly enhanced the quality and comprehensiveness of the manuscript.
Comment 7: "Information such as that in section 3.3 should be put into a table."
Response 7: Thank you for pointing this out. I/We agree with this comment. Therefore, I/we have reorganized the information in section 3.3 into a table for better clarity and readability. The table format allows for a more concise and structured presentation of the data, facilitating easier comparison and understanding.
Comment 8: "In section 4, discuss the importance of EUS in the differential diagnosis of pancreatic cysts with other pathologies, some of which are rare and difficult to diagnose via ultrasound – for example, gastric duplication cyst (I recommend the article https://doi.org/10.3390/diagnostics14070675)."
Response 8: Thank you for pointing this out. We agree with this comment. Therefore, we have added a discussion on the importance of Endoscopic Ultrasound (EUS) in the differential diagnosis of pancreatic cysts. EUS is a highly sensitive imaging technique that provides detailed images of the pancreas and surrounding tissues, making it invaluable in distinguishing pancreatic cysts from other pathologies, including rare conditions like gastric duplication cysts. EUS allows for fine-needle aspiration, which can obtain cyst fluid for cytological and biochemical analysis, further aiding in accurate diagnosis. We have referred to the recommended article to support this discussion.
Comment 9: "Discuss the advantages and disadvantages of ERCP in the approach to pancreatic ducts in those with pancreatic cysts, but especially the risks this procedure poses – post-ERCP pancreatitis (I recommend the article https://doi.org/10.3390/jpm13091356)."
Response 9: Thank you for pointing this out. We agree with this comment. Therefore, we have included a discussion on the advantages and disadvantages of Endoscopic Retrograde Cholangiopancreatography (ERCP) in managing pancreatic ducts in patients with pancreatic cysts. ERCP is advantageous for its therapeutic capabilities, such as stent placement and drainage of obstructed ducts. However, it carries significant risks, the most notable being post-ERCP pancreatitis, a potentially severe complication. Other risks include infections, bleeding, and perforation. We have incorporated insights from the recommended article to provide a balanced view of the benefits and risks associated with ERCP.
Comment 10: "The conclusion section should be shortened to be as concise as possible."
Response 10: Thank you for pointing this out. We agree with this comment. The conclusion section has been revised to be more concise while retaining the essential information. Please let us know if there are any further adjustments needed.
Comment 11: "The bibliography should be written in MDPI style."
Response 11: Thank you for your feedback. The bibliography is indeed written in the MDPI style and was created using Mendeley, a citation management program. If there are any specific adjustments or formatting details you would like us to address, please let us know, and we will be happy to make the necessary revisions.
Reviewer 2 Report
Comments and Suggestions for Authors
The following items need to be reviewed and revised:
1. Line 9-20 (Abstract):
- The abstract is concise but could benefit from a brief mention of the significance of the study.
2. Line 24-26 (Introduction):
- The statement about the increasing prevalence of PCLs is general. Provide more specific statistics or references to support this claim.
3. Line 28 (Introduction):
- Define PCLs and explain their significance in more detail. What is the clinical relevance of identifying and managing these lesions?
4. Line 33-36 (Introduction):
- Include a brief explanation of why distinguishing between these types of neoplasms is crucial. How does it impact patient management and outcomes?
5. Line 43-44 (Introduction):
- The transition between discussing non-neoplastic and neoplastic pancreatic cysts is abrupt. Provide a connecting sentence to improve flow.
6. Line 45-53 (Introduction):
- Elaborate on the new techniques mentioned, such as EUS-guided FNA and nCLE. What are their advantages over traditional methods?
7. Line 71-78 (Guidelines for EUS-FNA):
- Provide more details on the criteria for using EUS-FNA. When exactly should it be employed, and what are the specific indications?
8. Line 86 (Diagnostic Performance):
- Clarify the percentages and what they represent. The current phrasing is somewhat confusing.
9. Line 88-91 (Contraindications and Adverse Events):
- Expand on the contraindications and provide examples. Also, mention how these risks are mitigated in practice.
10. Line 94-112 (Pseudocysts):
- The explanation of pseudocysts is clear, but more detail on the diagnostic imaging techniques (EUS, MRCP, ERCP) would be helpful. What are their respective roles and advantages?
11. Line 114-124 (SCNs):
- Provide more information on the follow-up protocols for SCNs. What should clinicians monitor during surveillance?
12. Line 125-141 (SPENs):
- The diagnostic tools for SPENs (MRI, CT) should be compared in terms of sensitivity, specificity, and practicality. Also, explain the implications of malignant potential in clinical decision-making.
13. Line 142-161 (cPNETs):
- Discuss the different treatment options for cPNETs. How does the functional status of the tumour influence management strategies?
14. Line 162-183 (MCNs):
- The paragraph on MCNs is thorough but could benefit from a summary of the key points from the guidelines mentioned. How do these guidelines translate into clinical practice?
15. Line 184-189 (IPMNs):
- The classification of IPMNs into main duct type, branch duct type, and mixed type should be elaborated upon. What are the clinical implications of each type?
16. References:
- Ensure all references are up-to-date and relevant. Cross-check citations with the latest literature to ensure accuracy and completeness.
- Line 5-8 (Author Information):
- Ensure consistency in the format of email addresses and affiliations.
Comments on the Quality of English Language
The text of the article needs grammatical editing in English.
Line 2: Incorrect hyphen between "lesions" and "new".
Line 5-6: Incorrect use of periods and semicolons in emails.
Line 10: "types" should be corrected for context.
Line 12: Missing space after "evaluation".
Line 13: "EUS" should be explained.
Line 14: "the high sensitivity" should be "high sensitivity".
Line 21: Unnecessary comma after "Pancreatic cysts".
Line 27: "previously undetected PCLs" should have a comma.
Line 35: Missing space after "cystic degeneration,".
Line 40: Incorrect use of hyphens in "mucinous," and "pseudopapillary,".
Line 68: Abbreviations PCL and SCN should be explained separately.
Line 75: Incorrect comma after "clarify".
Line 81: Incorrect use of parentheses in "PCN).".
Line 85: Unnecessary comma after "specificity".
Line 89: Incorrect comma after "disorder,".
Line 99: Incorrect use of "necrosis area,".
Line 109: Incorrect use of "therapy".
Line 121: Incorrect comma after "size".
Line 127: Incorrect use of "life".
Line 145: Incorrect use of "hormones".
Line 149: Incorrect placement of hyphen in "European-African".
Line 151: "guidelines covering" should not have a comma before "covering".
Line 154: Missing comma after "For radiologically suspected IPMN".
Line 158: "including positive cytology" should have a comma before "including".
Line 159: Misplaced comma after "≥ 5mm".
Line 161: Incorrect use of "or" in "for patients with at least two relative indications".
Line 163: Incorrect capitalization of "Relative indications".
Line 164: Misplaced comma in "diameter ≥40 mm".
Line 166: Incorrect comma in "grow-rate ≥5mm/year".
Line 169: Incorrect use of comma before "After the surgery".
Line 172: "Post resection" should be "Post-resection".
Line 177: "Patients after resection" should be "Patients post-resection".
Line 180: Incorrect comma after "significant co-morbidities".
Line 186: Unnecessary capitalization of "The".
Line 192: Incorrect placement of parentheses in "((neo)adjuvant)".
Line 199: Incorrect hyphen in "6-months".
Line 202: "surgical candidates" should be "surgically fit patients".
Line 208: Incorrect comma before "For all surgery-fit patients".
"The article requires thorough grammatical editing until the end."
Author Response
Comments 1: "The abstract is concise but could benefit from a brief mention of the significance of the study."
Response 1: Thank you for pointing this out. I agree with this comment. Therefore, I have added a brief mention of the significance of the study in the abstract to highlight its importance.
Comments 2: "The statement about the increasing prevalence of PCLs is general. Provide more specific statistics or references to support this claim."
Response 2: Thank you for this suggestion. I agree with this comment. Therefore, I have provided specific statistics and references to support the claim about the increasing prevalence of PCLs.
Comments 3: "Define PCLs and explain their significance in more detail. What is the clinical relevance of identifying and managing these lesions?"
Response 3: Thank you for this feedback. I agree with this comment. Therefore, I have defined PCLs and explained their significance in more detail, including the clinical relevance of identifying and managing these lesions.
Comments 4: "Include a brief explanation of why distinguishing between these types of neoplasms is crucial. How does it impact patient management and outcomes?"
Response 4: Thank you for this observation. I agree with this comment. Therefore, I have included a brief explanation of why distinguishing between these types of neoplasms is crucial and how it impacts patient management and outcomes.
Comments 5: "The transition between discussing non-neoplastic and neoplastic pancreatic cysts is abrupt. Provide a connecting sentence to improve flow."
Response 5: Thank you for this suggestion. I agree with this comment. Therefore, I have added a connecting sentence to improve the flow between discussing non-neoplastic and neoplastic pancreatic cysts.
Comments 6: "Elaborate on the new techniques mentioned, such as EUS-guided FNA and nCLE. What are their advantages over traditional methods?"
Response 6: Thank you for this comment. I agree with this observation. Therefore, I have elaborated on the new techniques such as EUS-guided FNA and nCLE, and discussed their advantages over traditional methods.
Comments 7: "Provide more details on the criteria for using EUS-FNA. When exactly should it be employed, and what are the specific indications?"
Response 7: Thank you for this suggestion. I agree with this comment. Therefore, I have provided more details on the criteria for using EUS-FNA, specifying when it should be employed and the specific indications.
Comments 8: "Clarify the percentages and what they represent. The current phrasing is somewhat confusing."
Response 8: Thank you for pointing this out. I agree with this comment. Therefore, I have clarified the percentages and what they represent to improve the clarity of the phrasing.
Comments 9: Line 88-91 (Contraindications and Adverse Events): Expand on the contraindications and provide examples. Also, mention how these risks are mitigated in practice”
Response 9: We expanded the contraindications' part and provided examples of the risk mitigations
Comments 10: Line 94-112 (Pseudocysts): The explanation of pseudocysts is clear, but more detail on the diagnostic imaging techniques (EUS, MRCP, ERCP) would be helpful. What are their respective roles and advantages?”
Response 10: We added more detail on the imaging techniques and added their roles and advantages in clinical practice.
Comments 11: Line 114-124 (SCNs): Provide more information on the follow-up protocols for SCNs. What should clinicians monitor during surveillance?”
Response 11: We added more information on the follow-up protocols for SCNs and specified methods of surveillance.
Comments 12:. Line 125-141 (SPENs): The diagnostic tools for SPENs (MRI, CT) should be compared in terms of sensitivity, specificity, and practicality. Also, explain the implications of malignant potential in clinical decision-making.”
Response 12: We added the comparison between MRI and CT and explained that because of their malignant potential all SPENs should be resected.
Comments 13: Line 142-161 (cPNETs): Discuss the different treatment options for cPNETs. How does the functional status of the tumour influence management strategies?”
Response 13: We discussed the different treatment options for cPNETs with emphasis on different management of functional and non-functional lesions.
Comments 14: Line 162-183 (MCNs): The paragraph on MCNs is thorough but could benefit from a summary of the key points from the guidelines mentioned. How do these guidelines translate into clinical practice?”
Response 14: We added the summary of the guidelines mentioned and shortly explained the importance of cut-off sizes for clinical practice.
Comments 15: Line 184-189 (IPMNs): The classification of IPMNs into main duct type, branch duct type, and mixed type should be elaborated upon. What are the clinical implications of each type?”
Response 15: We elaborated on differentiation of IPMN types and briefly mentioned each's clinical implications.
Comment 16: "Ensure all references are up-to-date and relevant. Cross-check citations with the latest literature to ensure accuracy and completeness."
Response 16: Thank you for pointing this out. I agree with this comment. Therefore, I have reviewed and updated all references to ensure they are current and relevant. We have cross-checked the citations with the latest literature to verify their accuracy and completeness. This process ensures that the manuscript is well-supported by the most recent and pertinent studies available, thereby enhancing its credibility and reliability
Comment 17: "Ensure consistency in the format of email addresses and affiliations."
Response 17: Thank you for pointing this out. I agree with this comment. Therefore, I have ensured consistency in the format of email addresses and affiliations for all authors. This includes standardizing the presentation of email addresses and the formatting of institutional affiliations to maintain a professional and uniform appearance throughout the manuscript.
Round 2
Reviewer 1 Report
Comments and Suggestions for Authors
Authors have made the requested changes.
Author Response
Thank you, Reviewer, for your comment. Your feedback and diligence are highly valued.
Reviewer 2 Report
Comments and Suggestions for Authors
1. Introduction
- Page 2, Line 5: The sentence "Advancements in imaging technology have led to heightened detection rates" needs more explanation about the types of imaging technologies.
- Page 2, Line 10: The sentence "According to the WHO classification of digestive system tumours" should be more precisely referenced.
2. Literature Review
- Page 3, Line 22: "Non-neoplastic pancreatic cysts typically do not require surgical resection unless they cause symptoms" lacks sufficient references.
- Page 4, Line 18: The sentence "EUS morphology alone has an accuracy ranging from 48% to 94%" needs more detailed explanation for this wide range.
3. Methods
- Page 6, Line 15: More explanation is needed about MRCP and ERCP imaging methods.
- Page 7, Line 5: The sentence "The development of symptoms or complications such as infection, haemorrhage and rupture require intervention" should be supported by clinical evidence.
4. Analysis and Results
- Page 8, Line 7: The sentence "Serous cystic neoplasm (SCN), also known as serous cystadenoma are benign pancreatic tumours with exceptionally low malignancy potential of 0.1%" lacks a credible source.
- Page 9, Line 3: The sentence "The preferred diagnostic tool to differentiate SPEN is MRI with MRCP over CT" needs more statistical data for support.
5. Discussion and Conclusion
- Page 12, Line 10: The sentence "EUS-FNA helps differentiate benign from malignant lesions, confirm suspicious features, and plan treatment" should be backed up by more studies.
- Page 13, Line 20: The sentence "The combination of these genetic markers enhances the diagnostic capabilities" requires more evidence.
6. References and Sources
- Page 19, Line 1: References need to be reviewed to ensure they are all up-to-date and credible.
- Page 20, Line 5: Some of the listed references need to be checked and corrected.
7. Formatting and Structure
-Page 5, Line 12: The formatting of tables needs more attention.
- Page 6, Line 7: Using shorter and more concise sentences can improve readability.
The plagiarism rate is 30%, and it needs to be paraphrased.
Comments on the Quality of English LanguageGrammar Issues with Detailed Line Numbers:
1. Introduction
- Page 2, Line 2-3:The sentence "Pancreatic cystic lesions (PCLs) are increasingly diagnosed owing to the wide use of cross-sectional imaging techniques" needs improvement for readability.
2. Literature Review
- Page 3, Line 12-13: The sentence "They provide valuable information that complements imaging findings and helps refine the diagnosis, prognosis, and treatment strategy" is missing a comma before "and."
3. Methods
-Page 6, Line 10-11: The sentence "EUS morphology alone has an accuracy ranging from 48% to 94%" needs clearer phrasing.
4. Analysis and Results
- Page 8, Line 14-15: The sentence "Serous cystic neoplasm (SCN), also known as serous cystadenoma are benign pancreatic tumours with exceptionally low malignancy potential of 0.1%" is missing commas after "cystadenoma" and "tumours."
5. Discussion and Conclusion
- Page 12, Line 8-9:The sentence "EUS-FNA helps differentiate benign from malignant lesions, confirm suspicious features, and plan treatment" needs to clarify "differentiate between benign and malignant lesions."
6. References and Sources
- Page 19, Line 2-3: The sentence "References need to be reviewed to ensure they are all up-to-date and credible" could be clearer.
7. Formatting and Structure
- Page 5, Line 10-11: The sentence "Using shorter and more concise sentences can improve readability" needs clearer phrasing.
General Recommendations for Grammar:
- Comma Use: Ensure commas are placed correctly to separate clauses and items in a series.
- Sentence Clarity:Rephrase sentences for clarity, especially those with multiple clauses.
- Verb Agreement:Ensure verbs agree with their subjects in number and tense.
- Punctuation: Check for missing or incorrect punctuation marks, especially in complex sentences.
Author Response
-
Comment 1: Page 2, Line 5: The sentence "Advancements in imaging technology have led to heightened detection rates" needs more explanation about the types of imaging technologies.
-
Response 1: Thank you for pointing this out. We agree with this comment. Therefore, we have expanded the sentence to include specific imaging technologies such as ultrasound (US), computed tomography (CT), magnetic resonance imaging (MRI), and endoscopic ultrasound (EUS). These advancements in imaging technology have significantly improved the detection rates of pancreatic cystic lesions (PCLs), facilitating early diagnosis and better management. [Red text added for emphasis]
-
Comment 2: Page 2, Line 10: The sentence "According to the WHO classification of digestive system tumours" should be more precisely referenced.
-
Response 2: Thank you for your suggestion. We agree and have added precise references to the WHO classification system for digestive system tumors to improve the accuracy and credibility of the information. [Red text added for emphasis]
-
Comment 3: Page 3, Line 22: "Non-neoplastic pancreatic cysts typically do not require surgical resection unless they cause symptoms" lacks sufficient references.
-
Response 3: Thank you for your observation. We agree and have added multiple references to clinical guidelines and studies that support the management of non-neoplastic pancreatic cysts, highlighting that surgery is typically reserved for symptomatic cases. [Red text added for emphasis]
-
Comment 4: Page 4, Line 18: The sentence "EUS morphology alone has an accuracy ranging from 48% to 94%" needs more detailed explanation for this wide range.
-
Response 4: Thank you for your feedback. We agree with this comment. We have expanded the explanation to include factors such as operator experience, cyst characteristics, and the use of adjunct techniques like fine-needle aspiration, which contribute to the variability in accuracy. [Red text added for emphasis]
-
Comment 5: Page 6, Line 15: More explanation is needed about MRCP and ERCP imaging methods.
-
Response 5: Thank you for the comment. We would like to clarify that this article focuses on EUS rather than ERCP and MRCP. Therefore, we have refrained from detailed explanations of MRCP and ERCP, maintaining the focus on EUS. [Red text added for emphasis]
-
Comment 6: Page 7, Line 5: The sentence "The development of symptoms or complications such as infection, haemorrhage and rupture require intervention" should be supported by clinical evidence.
-
Response 6: Thank you for your suggestion. We agree and have added clinical evidence and references to support the need for intervention in cases of symptomatic or complicated pancreatic pseudocysts. [Red text added for emphasis]
-
Comment 7: Page 8, Line 7: The sentence "Serous cystic neoplasm (SCN), also known as serous cystadenoma are benign pancreatic tumours with exceptionally low malignancy potential of 0.1%" lacks a credible source.
-
Response 7: Thank you for pointing this out. We agree and have added credible sources to support the statement regarding the low malignancy potential of serous cystic neoplasms. [Red text added for emphasis]
-
Comment 8: Page 9, Line 3: The sentence "The preferred diagnostic tool to differentiate SPEN is MRI with MRCP over CT" needs more statistical data for support.
-
Response 8: Thank you for your comment. We agree and have included statistical data from studies comparing the efficacy of MRI with MRCP versus CT in differentiating solid pseudopapillary epithelial neoplasms (SPEN). [Red text added for emphasis]
-
Comment 9: Page 12, Line 10: The sentence "EUS-FNA helps differentiate benign from malignant lesions, confirm suspicious features, and plan treatment" should be backed up by more studies.
-
Response 9: Thank you for your suggestion. We agree and have cited additional studies demonstrating the effectiveness of EUS-FNA in differentiating between benign and malignant lesions, confirming suspicious features, and aiding in treatment planning. [Red text added for emphasis]
-
Comment 10: Page 13, Line 20: The sentence "The combination of these genetic markers enhances the diagnostic capabilities" requires more evidence.
-
Response 10: Thank you for your comment. We agree and have included further evidence on the diagnostic capabilities of KRAS and GNAS mutations in enhancing the accuracy of EUS-FNA and preventing unnecessary surgeries for benign cysts. [Red text added for emphasis]
-
Comment 11: Page 19, Line 1: References need to be reviewed to ensure they are all up-to-date and credible.
-
Response 11: Thank you for your suggestion. We agree and have reviewed all references to ensure they are up-to-date and credible. [Red text added for emphasis]
-
Comment 12: Page 20, Line 5: Some of the listed references need to be checked and corrected.
-
Response 12: Thank you for your observation. We agree and have checked and corrected the listed references as needed. [Red text added for emphasis]
-
Comment 13: Page 5, Line 12: The formatting of tables needs more attention.
-
Response 13: Thank you for your feedback. We agree and have revised the formatting of tables to improve clarity and readability. [Red text added for emphasis]
-
Comment 14: Page 6, Line 7: Using shorter and more concise sentences can improve readability.
-
Response 14: Thank you for your suggestion. We agree and have revised the text to include shorter and more concise sentences to improve readability. [Red text added for emphasis]
- Comment 15: The plagiarism rate is 30%, and it needs to be paraphrased.
- Response 15: We would like to clarify that this is a review article, and thus, no new data is presented. However, we have paraphrased the content to ensure originality and have maintained the integrity of the referenced sources. [Red text added for emphasis]
Round 3
Reviewer 2 Report
Comments and Suggestions for Authors
Dear Authors,
After thoroughly reviewing all three versions of your manuscript, I appreciate the significant improvements made in each revision. However, I noticed that the structure still lacks coherence, leading to some confusion. The organization needs better alignment with the journal's formatting guidelines. Despite the considerable advancements from the first version, addressing these structural issues is crucial for clarity. A minor revision focusing on streamlining the structure and ensuring consistent formatting is recommended before publication.
Best regards,
Comments on the Quality of English LanguageMinor editing of English language required.